# Exploring the Antioxidant Roles of Cysteine and Selenocysteine in Cellular Aging and Redox Regulation

**DOI:** 10.3390/biom15081115

**Published:** 2025-08-03

**Authors:** Marta Pace, Chiara Giorgi, Giorgia Lombardozzi, Annamaria Cimini, Vanessa Castelli, Michele d’Angelo

**Affiliations:** 1Department of Life, Health and Environmental Sciences, University of L’Aquila, 67100 L’Aquila, Italy; marta.pace@student.univaq.it (M.P.); chiara.giorgi2@graduate.univaq.it (C.G.); giorgia.lombardozzi@guest.univaq.it (G.L.); annamaria.cimini@univaq.it (A.C.); vanessa.castelli@univaq.it (V.C.); 2Sbarro Institute for Cancer Research and Molecular Medicine, Temple University, Philadelphia, PA 19122, USA

**Keywords:** aging, cysteine, selenocysteine, glutathione, oxidative stress, neurodegeneration

## Abstract

Aging is a complex, universal biological process characterized by the progressive and irreversible decline of physiological functions across multiple organ systems. This deterioration is primarily driven by cumulative cellular damage arising from both intrinsic and extrinsic stressors. The free radical theory of aging, first proposed by Denham Harman in 1956, highlights the role of reactive oxygen species (ROS), byproducts of normal metabolism, in driving oxidative stress and age-related degeneration. Emerging evidence emphasizes the importance of redox imbalance in the onset of neurodegenerative diseases and aging. Among the critical cellular defenses against oxidative stress are sulfur-containing amino acids, namely cysteine (Cys) and selenocysteine (Sec). Cysteine serves as a precursor for glutathione (GSH), a central intracellular antioxidant, while selenocysteine is incorporated into key antioxidant enzymes such as glutathione peroxidases (GPx) and thioredoxin reductases (TrxR). These molecules play pivotal roles in neutralizing ROS and maintaining redox homeostasis. This review aims to provide an updated and critical overview of the role of thiol-containing amino acids, specifically cysteine and selenocysteine, in the regulation of redox homeostasis during aging.

## 1. Introduction

Aging is a widespread biological phenomenon marked by the gradual and permanent loss of physiological function in multiple organ systems resulting from the cumulative impact of various internal and external stressors [1].

Aging research investigates the progressive loss of physiological functions that occur in organisms as they advance through adulthood. Currently, twelve key features are recognized as fundamental hallmarks of aging. These aging hallmarks include instability of the genome, shortening of telomeres, changes in epigenetic regulation, impairment of protein homeostasis, defective macroautophagy, disruptions in nutrient-sensing pathways, mitochondrial dysfunction, cellular senescence, depletion of stem cell reserves, altered cell-to-cell communication, persistent low-grade inflammation, and imbalances in the gut microbiota (dysbiosis) [2].

Among the twelve recognized hallmarks of aging, mitochondrial dysfunction is particularly significant. As organisms age, mitochondria exhibit increased vulnerability to structural alterations. These morphological changes impair their performance, largely due to oxidative damage caused by ROS accumulation, ultimately contributing to the overall aging process [3]. In fact, mitochondria represent a primary site for the generation of ROS [4]. ROS are known to play a key role in preserving tissue-specific physiological functions by reversibly modifying cysteine residues on proteins. However, aging disrupts ROS levels and redox signaling, which contributes to a progressive decline in tissue function [5,6].

In 1956, Denham Harman introduced the free radical theory of aging, suggesting that age-related degenerative processes are driven by the damaging actions of free radicals produced as byproducts of normal cellular metabolism [7]. Years later, in 1985, Helmut Sies described oxidative stress as a state where the equilibrium between prooxidants and antioxidants is disrupted, leading to damage to essential macromolecules [8].

The current perspective on oxidative stress recognizes two key dimensions: the traditional notion of macromolecular damage and the more recent discovery that disturbances in redox signaling and regulation contribute to aging-related diseases. Moreover, a distinction is made between physiological levels of oxidants, known as “eustress”, which support normal cellular functions and are essential for healthy longevity [9]. The concept of “eustress” was first introduced by Selye [10]. The term literally means “a beneficial or healthy response to a stress, associated with positive feelings” [11], and it is defined as “an optimal amount of stress” [12].

A growing body of research indicates that imbalances in redox homeostasis contribute to the onset of neurodegenerative disorders. Disturbances in cellular redox balance and signaling are closely linked not only to the progression of age-associated diseases but also to the fundamental mechanisms of aging [13]. Accumulating evidence over time has indicated that low concentrations of the oxidant hydrogen peroxide (H_2_O_2_) actively participate in redox-based cellular signaling mechanisms [14,15]. Moreover, it was discovered that the most prevalent low-molecular-weight thiol/disulfide pairs in plasma, namely glutathione/glutathione disulfide (GSH/GSSG) and cysteine/cystine (Cys/CySS), do not exist in a state of thermodynamic equilibrium [16].

Recent insights from the redox theory of aging suggest that aging is not solely driven by the accumulation of ROS, but rather by a progressive loss of redox resilience and adaptive capacity. This includes both increased oxidative damage and disruptions in redox signaling pathways, which compromise cellular homeostasis and stress responses [9]. Within this framework, thiol-containing amino acids such as cysteine and selenocysteine play pivotal roles: cysteine is essential for the synthesis of GSH, a central intracellular antioxidant that directly scavenges ROS and supports the activity of detoxifying enzymes [17], while selenocysteine is a critical component of key redox enzymes such as GPx and TrxR. The availability and proper regulation of these amino acids are therefore fundamental for maintaining redox balance and protecting against age-related functional decline and diseases [18].

Supplementation with cysteine has been shown to mitigate ROS-induced apoptosis and neurocognitive impairments, highlighting its protective role against oxidative stress [17]. Additional research supports the hypothesis that N-acetyl-L-cysteine (NAC) may exert neuroprotective effects in the aged rat brain [19]. Furthermore, dietary supplementation with selenocysteine has been observed to increase resistance to oxidative stress and extend lifespan in model organisms like *C. elegans*, suggesting its potential role in promoting longevity [20]. Recent evidence has indicated that administering a combination of glycine and N-acetylcysteine (GlyNAC) to older adults can enhance cellular antioxidant capacity by restoring GSH levels, while reducing oxidative stress, improving mitochondrial efficiency, and positively influencing biological markers associated with aging [21].

This review aims to provide an updated and critical overview of the role of thiol-containing amino acids, specifically cysteine and selenocysteine, in regulating redox homeostasis during the aging process. Given the multifactorial nature of aging and the complexity of redox regulation, identifying specific molecular mediators, such as thiol-containing amino acids, could provide a focused framework for better understanding and potentially modulating age-related decline.

## 2. Biochemical Properties of Cysteine and Selenocysteine

Cysteine and selenocysteine are structurally analogous amino acids, both of which are composed of a central carbon bonded to a hydrogen, a carboxyl group, an amino group, and a side chain containing a chalcogen atom, sulfur in cysteine (-CH_2_-SH) and selenium in selenocysteine (-CH_2_-SeH) [22]. Notably, the selenol group of selenocysteine is more nucleophilic and acidic (pKa ~5.2) compared to the thiol group in cysteine (pKa ~8.3), allowing selenocysteine to exist predominantly in its deprotonated and reactive thiolate form under physiological conditions [23]. This property significantly enhances its role in redox catalysis, especially in environments with fluctuating oxidative stress [24]. Many of the key distinctions between selenium and sulfur are derived from the general chemical trends observed when moving from lighter to heavier elements in the periodic table. Heavier atoms like selenium are more polarizable than their lighter counterparts, such as sulfur, which makes them “softer” in chemical terms. This increased polarizability facilitates faster electrophilic and nucleophilic substitution reactions that involve selenium. Additionally, chemical bonds involving selenium are generally weaker than those formed with sulfur, leading to more rapid bond cleavage. The weaker Se-X bonds have lower-energy antibonding orbitals, making them more reactive to electron donors. As a result, selenium-containing compounds exhibit greater electrophilicity across all oxidation states than their sulfur-based analogs [23]. These chemical properties are exploited in selenoproteins, where selenocysteine enhances redox activity beyond what cysteine can achieve, particularly in enzymes like GPx and TrxR, which require rapid and efficient peroxide reduction. Specifically, these enzymes (GPx and TrxR), which are essential in cellular antioxidant defense, incorporate Sec at their active sites. It is not surprising that most selenoproteins participate in redox reactions, as Sec is critical for their catalytic efficiency, whereas cysteine-containing analogs typically exhibit inferior enzymatic activity. Moreover, Sec-containing enzymes generally possess lower redox potentials than their cysteine-based counterparts, which enhances their function in the rapid detoxification of ROS, especially H_2_O_2_ and lipid hydroperoxides [23,24,25].

Structurally, the incorporation of selenocysteine is also unique at the translational level. While cysteine is directly coded by the standard genetic code (UGU/UGC), selenocysteine is encoded by the UGA codon, which is typically a stop codon, redefined in specific contexts via a highly conserved RNA secondary structure known as the selenocysteine insertion sequence (SECIS) element, which is a structural element of messenger RNA (mRNA) that is essential for the incorporation of Sec into proteins during translation, in conjunction with dedicated elongation factors and selenocysteine-specific tRNA (tRNA^Sec^). This complex machinery underscores the biological importance of selenocysteine and its evolutionary conservation across all domains of life, despite its synthetic complexity and higher metabolic cost [24].

In contrast, cysteine is abundant in redox-regulating proteins due to its ability to form disulfide bonds (-S-S-), acting as molecular switches in redox signaling and protein folding [26].

Cysteine residues play a pivotal role in redox-regulating proteins by undergoing various oxidative modifications in response to ROS. Under mild oxidative conditions, the thiol group (-SH) can be reversibly oxidized to form sulfenic acid (-SOH), which acts as a transient redox switch modulating protein function. Under more intense oxidative stress, further irreversible oxidations may occur, producing sulfinic (-SO_2_H) or sulfonic acids (-SO_3_H), which are typically associated with loss of enzymatic activity. These dynamic modifications contribute to redox regulation in proteins such as protein disulfide isomerases (PDIs) and peroxiredoxins (Prxs) and highlight the critical sensing role of cysteine in oxidative signaling networks [27].

In summary (Figure 1), although cysteine and selenocysteine share a structural backbone, their chemical properties diverge significantly due to the chalcogen substitution [22]. These differences translate into unique biological roles: cysteine primarily supports structural stability and redox buffering, whereas selenocysteine provides enzymes with superior catalytic potential in antioxidant defense. The complementary functions of cysteine and selenocysteine underscore the necessity of both amino acids in maintaining cellular redox homeostasis, particularly under conditions of oxidative stress and aging [9,18,24,26].

## 3. Metabolism and the Synthesis of Selenocysteine (UGA Codon, SECIS Element)

Currently, 22 amino acids are known to participate in protein biosynthesis. Among them, selenocysteine, recognized as the 21st amino acid, is classified as a non-standard amino acid. It is uniquely encoded by the UGA codon, which typically signals the termination of translation. Due to this dual role of the UGA codon, the incorporation of selenocysteine into selenoproteins necessitates specialized translational and biosynthetic mechanisms within the cell [28].

Selenoproteins that contain selenocysteine play diverse roles in promoting human health and hold a substantial promise for biomedical and commercial applications. As a critical catalytic residue in enzymes such as GPx, TrxR, and selenoprotein P (SELENOP), selenocysteine contributes to a range of physiological processes, including cellular protection against aging, the modulation of inflammatory responses, and defense against cancer [28].

The biosynthetic route of selenocysteine has only been elucidated in recent years through integrated studies involving comparative genomics, molecular biology, and structural analysis. What sets Sec apart from all other amino acids is that in eukaryotes, it is uniquely synthesized directly on its own specialized tRNA, known as tRNA^[Ser]Sec^, which works in collaboration with a unique RNA structure in the 3′ untranslated region of the mRNA known as the Sec Insertion Sequence (SECIS) element, and several essential protein cofactors. This tRNA is initially charged with a serine residue by the enzyme seryl-tRNA synthetase (SerRS), forming seryl-tRNA^[Ser]Sec^, which then serves as the precursor scaffold for the subsequent formation of selenocysteine [24]. The tRNA^[Ser]Sec^ gene, known as *Trsp*, was first identified and sequenced in chickens [29].

There are several features of tRNA^[Ser]Sec^ that distinguish it from other tRNAs [24].

The first step in selenocysteine biosynthesis in both eubacteria and eukaryotes involves the aminoacylation of tRNA^[Ser]Sec^ with serine, which serves as the molecular backbone for subsequent selenocysteine formation [24]. The initial aminoacylation of tRNA^[Ser]Sec^ with serine by SerRS indicates that this tRNA contains identity elements recognized by SerRS for serine, but not for selenocysteine. These recognition elements are located in the discriminator base and the extended variable arm, both of which are critical for efficient aminoacylation by SerRS [30,31].

The spatial arrangement of the extended variable arm in the three-dimensional structure of tRNA^[Ser]Sec^ closely resembles that of canonical tRNA^Ser^, suggesting that it is likely recognized by SerRS through a similar molecular interaction mechanism [32].

The transformation of the serine residue attached to tRNA^[Ser]Sec^ into selenocysteinyl-tRNA^[Ser]Sec^ is catalyzed by the enzyme selenocysteine synthase (SecS), which utilizes selenophosphate, the biologically active form of selenium, as the selenium donor to complete the synthesis of Sec-tRNA^[Ser]Sec^ [33].

The incorporation of Sec into nascent polypeptides occurs co-translationally, and it is directed by in-frame UGA codons found within selenoprotein mRNAs. When the ribosome encounters a UGA codon, the Sec incorporation machinery engages with the standard translation apparatus to reinterpret UGA as a codon for Sec, thereby avoiding premature termination. The SECIS element plays a pivotal role in this recoding event, enabling the recruitment of the Sec-tRNA^[Ser]Sec^, whose anticodon forms a base pair with the UGA codon. In eukaryotes, the efficient decoding of UGA as selenocysteine requires at least two trans-acting protein factors: SECIS-binding protein 2 (SBP2), which interacts directly with the SECIS element, and the Sec-specific elongation factor (eEFSec), which delivers Sec-tRNA^[Ser]Sec^ to the ribosome [24].

## 4. Metabolic Pathways and Dietary Sources of Cysteine and Selenocysteine

Having established the structural and biochemical distinctions between cysteine and selenocysteine, we now explore their metabolic pathways and dietary sources. Cysteine is typically classified as a non-essential amino acid, as it can be endogenously synthesized from methionine via the transsulfuration pathway. However, under specific physiological or pathological conditions, such as oxidative stress or inflammation, it may become conditionally essential because sulfur-containing amino acids play a crucial role in physiological processes, as they serve as precursors to essential biomolecules. For instance, cysteine is a key precursor to glutathione, which is involved in cellular defense against oxidative stress.

During these states, the demand for cysteine increases substantially, primarily due to its critical role in glutathione biosynthesis, the cell’s principal intracellular antioxidant, as well as its incorporation into acute-phase proteins. When cysteine utilization surpasses the rate of endogenous synthesis, dietary intake becomes necessary to meet metabolic needs [34].

The transsulfuration pathway is a key component of cellular sulfur metabolism and redox homeostasis. In mammals, this biochemical route facilitates the conversion of homocysteine, an intermediate derived from the metabolism of dietary methionine, into cysteine. This transformation proceeds through the intermediate compound cystathionine. Initially, homocysteine is converted to cystathionine through the enzymatic action of cystathionine β-synthase (CBS). Subsequently, cystathionine is cleaved by cystathionine γ-lyase (CSE), resulting in the formation of cysteine. Notably, this pathway represents the sole endogenous mechanism for cysteine biosynthesis in mammalian cells [35].

Selenocysteine is synthesized co-translationally in the human body. As previously specified, this unique process involves the recording of the UGA stop codon to incorporate Sec into selenoproteins. The biosynthesis pathway requires dietary selenium, which is metabolized into selenophosphate, the active selenium donor essential for Sec formation. This mechanism is distinct from the direct incorporation of amino acids from dietary proteins [36].

Selenium is an indispensable micronutrient involved in maintaining normal physiological functions in humans. It is utilized through its incorporation into 25 characterized selenoproteins, which contribute to diverse biological roles, including redox regulation, protection against oxidative stress, and cancer development. The primary source of selenium for humans is the diet, with selenium-enriched products typically originating from plant sources (such as tea, rice, and garlic), animal-derived foods (including meat, eggs, and dairy), and microbial sources (like yeast and fungi) [37].

A further important factor is that plants serve as the principal entry point of selenium into the food chain. In contrast, selenium intake from drinking water is generally insignificant [38]. The selenium content of food is primarily influenced by soil-related factors. Although the total selenium concentration in soil is determined by the underlying source material, its bioavailability to plants depends on several variables, including soil pH, redox potential, organic matter content, the presence of competing anions such as sulfate, microbial activity, soil texture and compaction, mineral composition, temperature, irrigation, and rainfall during the growing season. Variations in soil moisture and pH caused by climatic fluctuations also play a significant role in selenium uptake by food plants [39].

Another important factor influencing selenium intake from the diet is the method of food preparation. According to a previous study [40], cooking methods can lead to a significant reduction in the selenium content of many foods. Researchers have demonstrated that selenium-rich vegetables, such as asparagus and mushrooms, can lose up to 40% of their selenium content during boiling due to water-mediated leaching. Additional studies have reported that cooking can cause selenium losses of up to 50% in vegetables and dairy products, particularly when ingredients like salt or acidic components (e.g., vinegar) are added. In contrast, frying tends to preserve a greater proportion of selenium, resulting in lower nutrient loss [40,41,42]. Therefore, the method of food preparation should be considered when assessing both the quantity and bioavailability of dietary selenium intake.

## 5. Redox Signaling Pathways

At physiological concentrations, ROS function as essential signaling molecules involved in a wide range of cellular processes, such as cell growth, proliferation, differentiation, and apoptosis, immune regulation, and adaptation to stress [43]. Recent reviews have highlighted the fact that ROS act as secondary messengers, modulating the activity of numerous enzymes, transcription factors, such as NF-kB, Nrf2, p53, and MAPK, and signaling cascades. These reactive molecules thus serve crucial regulatory functions in physiological contexts, while their overproduction leads to oxidative damage and pathology. Appreciating this dual role of ROS, as both signaling intermediates and potential mediators of molecular injury, is essential for dissecting the interplay between metabolism, redox homeostasis, and gene regulation during aging [44]. This review focuses on the major ROS-regulated signaling pathways.

The Activator Protein-1 (AP-1) family includes a heterogeneous group of transcription factors belonging to the basic leucine zipper (bZIP) class, which plays a crucial role in regulating cellular responses to a wide range of external stimuli, including ROS. These bZIP transcription factors form one of the most extensive dimerization networks across all eukaryotic organisms [45]. The bZIP transcription factors are characterized by a C-terminal leucine zipper domain, allowing them to build up homodimers and heterodimers, which have the ability to bind DNA. An amino acid-rich domain located next to the leucine zipper region enables this DNA binding. AP-1 transcription factors, both homodimeric and heterodimeric complexes, are composed of proteins from the Jun (c-Jun, JunB, and JunD), Fos (c-Fos, FosB, Fra-1, and Fra-2), Maf (including c-Maf, MafA, MafB, MafG/F/K, and Nrl), and ATF (such as ATF2, ATF3, B-ATF, JDP1, and JDP2) subfamilies [46]. AP-1 proteins are essential regulators of key cellular functions, such as cell growth, specialization, and programmed cell death [47]. AP-1 transcription factors are stimulated by a broad spectrum of external signals, including growth factors, cytokines, neurotransmitters, hormones, and environmental stressors, such as toxic agents and ultraviolet (UV) radiation. Notably, multiple studies have demonstrated that AP-1 exhibits highly conserved redox-sensitive behavior across species, from yeast to mammals, where its activation by extracellular cues is dependent on the presence of ROS [48,49]. The ability of c-Fos and c-Jun to bind DNA is influenced by the redox status of specific conserved cysteine residues [50]. Furthermore, ROS signaling regulates AP-1 in an indirect way. In fact, the transcription factors of the AP-1 family are phosphorylated through mitogen-activated protein kinase (MAPK) pathways, which are activated by ROS, resulting in increased transcriptional activity [48,51]. Accurate redox regulation of AP-1 activity is crucial for preserving cellular equilibrium, and its dysfunction has been associated with a range of pathological states, including tumor development, chronic inflammation, and neurodegenerative disorders [44]. In addition, the involvement of AP-1 in inflammaging pathways is of the utmost importance [52].

Nuclear factor-kappa B (NF-kB) is a widely studied family of transcription factors, including NF-kB1, NF-kB2, p65/RelA, c-Rel, and RelB, that regulate the expression of numerous genes involved in cell survival, growth, inflammatory responses, and immune system activity. A defining feature of all NF-kB proteins is the presence of a Rel homology domain (RHD), which is crucial for their ability to form homo- or heterodimers and to bind specific DNA sequences, thereby enabling their transcriptional regulatory functions. The canonical NF-kB signaling pathway is mainly triggered by the activation of proinflammatory receptors, particularly members of the TNF receptor superfamily, which leads to the nuclear translocation of NF-kB dimers and their binding to specific gene promoters [53]. In addition, NF-kB proteins are key regulators of both antioxidant and pro-oxidant gene expression, contributing to the cellular defense mechanisms against oxidative stress induced by ROS [54]. At physiological levels, ROS function as secondary messengers that actively contribute to signal transduction pathways culminating in NF-kB activation, while various antioxidants are capable of effectively inhibiting this activation [55]. This signaling pathway is characterized by a sequence of events in which ROS directly influence and activate the IkB kinase (IKK) complex, which in turn phosphorylates the NF-kB inhibitory protein IkB. This phosphorylation marks IkB for degradation via the ubiquitin–proteasome system, thereby releasing NF-kB dimers from their inactive cytoplasmic complexes. Once disengaged from their inhibitory complexes, these dimers translocate into the nucleus, where they initiate the transcription of target genes [56]. In contrast, when ROS levels rise above physiological levels, such as during exposure to pathogens, proinflammatory cytokines, or environmental stressors, or as a result of aging, they can negatively impact the regulation of NF-kB signaling, potentially leading to dysregulated inflammatory and stress responses [57]. In this condition, ROS levels lead to prolonged activation of the NF-kB pathway by disrupting the negative feedback mechanisms that usually regulate its activity. Additionally, ROS can directly alter key cysteine residues in proteins involved in the NF-kB signaling cascade, such as IKK, thereby contributing to its sustained activation. A further mechanism involves DNA damage triggered by ROS because it can indirectly activate the NF-kB pathway. DNA damage sensors (ATM and ATR kinases) detect strand breaks and other genomic lesions and respond by phosphorylating NF-kB essential modulator (NEMO), a regulatory component of the IKK complex. This modification facilitates the activation of IKK, thereby initiating NF-kB signaling and establishing a connection between genotoxic stress and inflammatory responses [58].

p53, which is widely recognized as the “guardian of the genome”, plays a pivotal role in preserving genomic stability and coordinating cellular responses to various stressors [59]. In the context of cellular aging and DNA damage, p53 becomes activated, governing the progression of the cell cycle from the G1 phase to the S phase [60]. ROS closely regulate the function of p53, functioning as both upstream activators and dynamic controllers of this pathway [61]. Consequently, cellular stress is recognized as a driving factor in the aging process and contributes to epigenetic alterations. In particular, chromatin remodeling, which is triggered by exposure to histone deacetylase inhibitors, can promote cellular senescence through the activation of p21, a cyclin-dependent kinase inhibitor regulated by p53, which links DNA damage to cell cycle arrest [62,63,64,65]. Enhanced resistance to oxidative stress has been linked to delayed aging in mammals and a reduced incidence of tumor development, suggesting the suppression of downstream effectors within the p53 signaling cascade. This regulatory mechanism was investigated in mice lacking the p66 gene, a mutation that conferred both cellular and systemic protection against oxidative damage and contributed to lifespan extension [60]. A transcriptional network comprising around 200 genes, which is repressed by p53 and involved in promoting mitotic progression and delaying aging, was identified. These genes exhibited selective downregulation both in vitro in fibroblast cultures exposed to oxidative stress and in vivo during physiological aging. This selective repression was attributed to the absence of the p66 gene and the activation of the p44/p53 isoform (also known as Delta40p53), which has been implicated in premature aging by mitosis inhibition following protein damage. The deletion of p66 was associated with delayed aging and an increased lifespan in transgenic mice overexpressing p44/p53 [60,65].

The Keap1-Nrf2-ARE signaling pathway is a well-characterized regulatory mechanism that is essential for maintaining cellular redox balance and protecting cells from endogenous and exogenous stressors [66]. ROS are key regulators of this pathway, modulating the dynamic interplay between Nrf2 activation and its suppression by Keap1 [67]. Under basal conditions, Keap1 functions as an adaptor protein for the Cullin3-based E3 ubiquitin ligase complex. In the cytoplasm, it binds to Nrf2 and facilitates its ubiquitination, targeting it for proteasomal degradation. This mechanism ensures that Nrf2 levels remain low in the absence of stress. However, when intracellular ROS levels increase, particularly in the presence of H_2_O_2_, they act as redox signals that alter specific cysteine residues on Keap1. These oxidative modifications induce conformational changes in Keap1, impairing its ability to promote Nrf2 ubiquitination. As a result, Nrf2 becomes stabilized, and it translocates to the nucleus. There, it forms heterodimers with small Maf proteins and binds to Antioxidant Response Elements (AREs) in the promoter regions of target genes. This binding activates the transcription of a broad range of antioxidant and detoxification enzymes, including NADPH quinone oxidoreductase 1 (NQO1), heme oxygenase-1 (HO-1), and glutathione S-transferases (GTSs). Collectively, these proteins work to counteract oxidative stress and electrophilic damage by neutralizing reactive molecules and reinforcing the cell’s antioxidant defense system [44,68].

Given their central role in antioxidant defense and redox-sensitive signaling cascades, cysteine and selenocysteine emerge as crucial modulators of these ROS-mediated pathways, particularly during aging, where maintaining redox homeostasis becomes increasingly dependent on finely regulated thiol- and selenol-based mechanisms.

## 6. Epigenetic Modification During Aging

A growing body of research across both invertebrate and vertebrate models, as well as tissue and cell-based studies, supports a strong connection between epigenetic regulation and the aging process. In mammals, aging is accompanied by widespread and site-specific alterations in DNA methylation patterns, a global reduction in histone content, and extensive chromatin restructuring. Moreover, RNA modifications and the activity of non-coding RNAs contribute significantly to cellular senescence through post-transcriptional gene regulation. Understanding how these epigenetic factors influence aging at the molecular level may reveal potential therapeutic targets for slowing the aging process and promoting tissue rejuvenation [69].

DNA methylation typically takes place at cytosine residues within CpG dinucleotides, leading to the formation of 5-methylcytosine (5-mC), with 60% to 90% of these CpG sites being methylated in the mammalian genome. During aging, however, a global reduction in DNA methylation is commonly observed. This hypomethylation correlates with increased expression of genes involved in energy metabolism and oxidative stress response, particularly in the skeletal muscle of older individuals [70,71]. DNA methyltransferases (DNMTs), including DNMT1, DNMT3A, and DNMT3B, catalyze the addition of methyl groups to cytosine bases in DNA, a process that typically leads to the repression of gene expression [72,73]. DNMT1 expression declines with advancing age, contributing to a global decrease in DNA methylation. Certain DNMT1 mutants associated with the degeneration of specific central and peripheral neurons have been observed to relocate from the nucleus to the cytoplasm, where they aggregate into aggresomes and lose their ability to interact with heterochromatic regions [74]. Conversely, the expression of DNMT3A and DNMT3B rises with age, promoting de novo methylation at CpG island regions in mammalian genomes as aging progresses [75,76]. The removal of DNA methylation marks is mediated by the ten-eleven translocation (TET) family of enzymes [77]. Clinical studies in elderly individuals have shown that mutations in TET2 or DNMT3A are linked to elevated levels of pro-inflammatory cytokines and persistent inflammation, conditions commonly associated with traditional cardiovascular diseases (CVD) [78]. Several investigations have shown that selenium levels or supplementation can influence both global and gene-specific DNA methylation patterns and impact the expression and enzymatic activity of DNA methyltransferases (DNMTs). Arai et al. exposed murine embryonic stem cells to a physiologically relevant, non-toxic concentration of selenium comparable to levels found in maternal serum. This treatment led to reversible modifications to heterochromatin organization and induced gene-specific changes in DNA methylation. Importantly, these epigenetic changes did not impair the cell’s ability to differentiate into embryoid bodies [79]. The alterations in chromatin architecture reported by Arai et al. may be attributed to variations in overall DNA methylation levels between selenium-supplemented cells and those experiencing selenium deficiency [80]. Research conducted on rodents [81,82,83,84] and cellular models [85] has demonstrated that dietary selenium intake can influence global DNA methylation levels. However, the findings across rodent studies have been inconsistent, with some reporting increased methylation, while others have observed reductions, even when using similar selenium dietary regimens. For example, selenium deficiency was associated with reduced DNA methylation in rat livers [82,83] and colons [83], whereas a study by Zeng et al. found elevated methylation levels in these same tissues when rats were provided with selenium-enriched diets compared to those receiving adequate or deficient levels. Zeng et al. suggested that variations in the animal strains used and the composition of the baseline diets could influence how selenium exerts its effects [81].

A decline in DNA methylation is commonly observed with aging in various human and murine tissues, as well as in cultured cells [86,87,88,89]. In individuals over the age of 100, CD4+ T cells exhibit a reduced level of genome-wide DNA methylation when compared to those of newborns [87]. A reduction in 5-methylcytosine (5-mc) levels has been detected in multiple organs, such as the brain, liver, and intestinal mucosa, when comparing young and aged mice, and this decline compromises cellular physiological functions in older animals [86]. In addition, certain CpG sites exhibit greater methylation variability as age progresses; these are referred to as age-associated variability methylated positions (aVMPs) [90,91]. Moreover, certain CpG sites exhibit consistent age-related methylation changes and are known as age-associated differentially methylated positions (aDMPs) [92]. The level of methylation at aDMPs has been shown to decline with age across six mammalian species, including humans [89]. Therefore, age-related changes in DNA methylation involve various CpG sites, such as aVMPs and aDMPs, which can be analyzed to estimate an individual’s epigenetic age [69].

Post-translational modifications that occur on histone proteins can either promote or repress gene activity, thereby influencing the aging process. These histone modifications encompass various types, such as methylation, acetylation, phosphorylation, ubiquitination, and ADP-ribosylation, among others [93]. Among the various histone modifications, the methylation and acetylation of lysine residues are the most extensively investigated, and they are recognized for their impact on the regulation of aging [69]. Earlier research has demonstrated that H3K4me3, a histone mark linked to active gene transcription, significantly contributes to the regulation of aging and lifespan by influencing the expression of genes involved in age-related pathways [94,95]. In the prefrontal cortex of elderly individuals (over 60 years), neurons show a reduction in H3K4me3 levels at 556 genes and an increase at 101 genes when compared to neurons from infants (under 1 year of age) [96]. Conversely, in a mouse model of Alzheimer’s disease (AD), elevated levels of H3K4me3 and its associated methyltransferase enzymes have been detected in the prefrontal cortex, an area that is critically affected by the disease. Treatment with an inhibitor targeting H3K4 methyltransferases has been shown to restore functional activity in this brain region [97]. In contrast to histone methylation, the link between overall histone acetylation and lifespan is more clearly defined. This modification, facilitated by lysine acetyltransferases (KATs), is typically enriched in regions of active transcription. Histone deacetylases (HDACs) generally act as transcriptional repressors and, in coordination with KATs, play a vital role in regulating aging and longevity. Among HDACs, sirtuins, classified as class III HDACs, contribute to genomic stability and modulate lysine deacetylation in an NAD^+^-dependent fashion [98,99]. Within the sirtuin family, SIRT1 levels have been found to decline with age in various tissues of both humans and mice, including the liver, heart, kidney, lung, and brain [100,101]. SIRT6 acts as an NAD^+^-dependent deacetylase that targets H3K9, playing a role in the regulation of telomeric chromatin. Its increased expression promotes the longevity of nucleus pulposus cells (specialized cells found within the nucleus pulposus, the gel-like center of the intervertebral disc (IVD) in the spine) in both rats and humans by suppressing cellular aging [102]. Selenium and other micronutrients have been found to either trigger or correlate with alterations in histone modifications, potentially influencing health outcomes. These nutrients may affect histone marks by altering the activity or expression of enzymes responsible for histone modification, or by impacting the availability of substrates necessary for these processes. Due to the broad diversity of histone modifications and the many enzymes involved, the regulatory landscape is even more intricate than that of DNA methylation. Additionally, interactions between DNA methylation and histone modifications create a highly interconnected and complex epigenetic regulatory network [103].

Chromatin is a dynamic and adaptable structure made up of DNA and histone proteins, and it can be organized into either tightly packed heterochromatin or poorly arranged euchromatin. Its fundamental unit is the nucleosome, which consists of DNA wrapped around an octamer of histones. This octamer is formed by a central H3-H4 tetramer, by two H2A-H2B dimers [104]. Extensive changes in chromatin architecture have been observed during cellular senescence, including variations in histone composition and modifications, as well as shifts in chromatin compartment organization and topologically associating domains (TADs) [105,106,107]. Of particular interest is the advancement of technologies like single-cell omics sequencing, which offer greater resolution in characterizing epigenetic features throughout the aging process and enable deeper investigation into cellular heterogeneity in aged tissues. Moreover, the development of spatiotemporal transcriptomic maps across various mammalian organs provides valuable insight into age-associated intercellular and intratissue interactions, potentially guiding the creation of more targeted and effective therapies for aging and age-related diseases [69].

Cysteine and selenocysteine significantly contribute to the regulation of epigenetic mechanisms in the aging brain, primarily through their roles in redox balance and metabolic integration. Selenocysteine-containing proteins, such as glutathione peroxidases and thioredoxin reductases, influence DNA methylation, histone modifications, and non-coding RNA expression by modulating the cellular redox environment. In hematopoietic stem cells, disruption of selenoprotein synthesis leads to epigenetic reprogramming, ferroptosis, and altered lineage commitment.

Moreover, selenoproteins, which incorporate selenocysteine, are essential for maintaining neuronal redox homeostasis and have been shown to influence DNA methylation, histone modifications, and non-coding RNA expressions [108]. These effects are particularly relevant in neurodegenerative conditions such as Alzheimer’s disease, where altered selenoprotein expression correlates with epigenetic dysregulation and neuronal decline [109]. Cysteine, as a precursor to glutathione, supports the activity of redox-sensitive epigenetic enzymes and contributes to one-carbon metabolism, which is crucial for methylation reactions. In the aging brain, shifts in cysteine metabolism can affect the availability of methyl donors and the redox environment, thereby influencing gene expression patterns associated with cognitive decline and neuroinflammation [110]. Together, these amino acids act as metabolic–epigenetic integrators, linking oxidative stress, nutrient sensing, and gene regulators in the context of aging processes and disease.

## 7. Gene Editing Strategies in Aging and Redox Balance

Genetic alterations that affect pathways related to DNA repair, cellular metabolism, and inflammatory responses can accelerate aging by disrupting normal cellular functions and facilitating the gradual accumulation of molecular damage. In addition, epigenetic modification, such as changes in DNA methylation and histone marks, can influence aging by reshaping gene expression profiles and altering cellular activity over time [111]. Gaining insights into the molecular and cellular mechanisms that underlie aging is essential for developing strategies aimed at promoting healthy aging and preventing age-associated diseases [112]. Currently, the CRISPR/Cas9 system, which is based on clustered regularly interspaced short palindromic repeats and the Cas9 endonuclease, has emerged as a highly promising genome editing technology, attracting considerable interest in biomedical research [112]. Researchers have achieved substantial advancements in understanding the molecular basis of aging using CRISPR/Cas9 technology, which has revolutionized the fields of genetics and molecular biology [113]. As previously mentioned, aging is characterized by a progressive decline in cellular function, which contributes to the development of various health conditions, including cardiovascular disease, cancer, and neurodegenerative disorders like AD. A significant study investigated a mouse model of Hutchinson–Gilford progeria syndrome, a rare genetic condition that causes premature aging. These mice displayed multiple aging-related symptoms, including genomic instability, impaired heart function, and a notably reduced lifespan [114]. The research team designed an innovative CRISPR/Cas9-based gene-editing strategy aimed at counteracting the accelerated aging phenotype observed in the progeria mouse model. Their approach focused on the LMNA gene, which encodes two closely related proteins, Lamin A and Lamin C. In Hutchinson–Gilford progeria syndrome, a mutation leads to the preferential production of a toxic variant known as progerin. Through precise genome editing, the researchers were able to suppress the expression of progerin and mitigate its damaging effects, providing new insights into the molecular mechanisms of aging [115]. A separate investigation into aging employed a genome-wide CRISPR/Cas9 screening approach to uncover multiple previously unrecognized genes that contribute to the induction of cellular senescence in humans [116]. CRISPR/Cas9 technology has the potential to specifically modulate cysteine/selenocysteine-related genes, thereby modulating fundamental factors involved in the aging process. For instance, a recent study developed an innovative strategy, employing CRISPR/Cas9 encapsulated within viral-like particles (VLPs) to selectively disrupt the gene encoding the tRNA^[Ser]Sec^ in human cell lines. This targeted deletion resulted in a substantial downregulation of selenoprotein expression, with levels of key selenoproteins such as GPX1 and GPX4 reduced by up to 90% in certain cellular models [117]. Complementarily, a genome-wide CRISPR-Cas9 dropout screening in human acute myeloid leukemia (AML) cells identified SEPHS2, a gene critical for the selenocysteine biosynthetic pathway. The disruption of SEPHS2 led to impaired GPX4 synthesis, elevated intracellular ROS, and increased sensitivity to ferroptosis [118]. These findings highlight the utility of CRISPR-based technologies in precisely modulating components of the selenoprotein biosynthetic machinery and demonstrate their potential to uncover redox vulnerabilities. Notably, such approaches may offer valuable tools for investigating the regulation of selenoprotein expression in aging contexts, where oxidative stress and redox imbalance are key drivers of cellular dysfunction. By directly targeting core elements of the selenocysteine incorporation pathway, this strategy provides a promising framework for exploring how the modulation of selenoprotein synthesis influences age-related physiological decline.

## 8. Cysteine, Selenocysteine, and Redox Regulation

Among sulfur-containing amino acids, cysteine stands out for its dual role as a building block of proteins and a key modulator of cellular redox balance. Cysteine is a sulfur-containing amino acid distinguished by its functional versatility, which is primarily attributed to its highly reactive thiol (–SH) group. Although it is among the least abundant amino acids, cysteine residues are frequently conserved in critical regions of proteins, where they play essential roles in catalysis, regulation, and ligand binding. The unique reactivity of the thiol or thiolate moiety underlies the cysteine’s capacity to act as a potent nucleophile, bind metal ions with high affinity, and form stabilizing disulfide bonds. It is also susceptible to oxidative stress due to the high reactivity of its thiol group [119].

As mentioned above, cysteine plays a central structural and functional role in GSH, which is a vital intracellular tripeptide composed of glutamic acid, cysteine, and glycine (γ-glutamyl–cysteinyl–glycine).

GSH is present in the cytosol of all mammalian tissues at concentrations typically ranging from 1 to 10 millimolar [120].

Sulfur and selenium share comparable physicochemical characteristics due to their classification as chalcogens, enabling them to participate in thiol-disulfide exchange reactions through Cys and Sec, respectively. Despite these similarities, Sec exhibits greater chemical reactivity than Cys under physiological conditions, largely due to its lower pKa (~5.2 versus ~8.3 for Cys). This lower pKa allows Sec to remain in its deprotonated, nucleophilic form at neutral pH without requiring stabilization by nearby charged groups, thereby enhancing its catalytic performance [18].

A defining characteristic shared by all selenoproteins is the inclusion of Sec residues within their amino acid sequences. In most cases, Sec is positioned at the active site of the enzyme, where it has evolved to facilitate redox-based catalytic functions [22].

There are 25 known genes that code for selenoproteins [121,122]. In humans, the selenoprotein family comprises a diverse group of proteins, including the glutathione peroxidases (GPX1 to GPX4 and GPX6), the thioredoxin reductases (TXNRD1 and TXNRD2), and thioredoxin–glutathione reductase (TXNRD3). Other members include the iodothyronine deiodinases (DIO1, DIO2, and DIO3), selenophosphate synthetase (SEPHS2), the 15-kDa selenoprotein (SELENOF), and various proteins designated by letter, such as SELENOH, SELENOI, SELENOK, SELENOM, SELENON, SELENOO, and SELENOP. Additionally, the family includes methionine sulfoxide reductase B1 (MSRB1) and selenoproteins S (SELENOS), T (SELENOT), V (SELENOV), and W (SELENOW) [123]. Selenoproteins are involved in the oxidative stress process, which is also linked to the aging process.

GPxs perform diverse physiological roles across organisms, particularly in modulating H_2_O_2_ signaling, neutralizing hydroperoxides, and preserving the redox equilibrium within cells. Among them, GPx1 is the most prevalent selenoprotein found in mammals; moreover, GPx1 is typically the dominant cytoplasmic form in most tissues [124]. This cytosolic enzyme facilitates the reduction of H_2_O_2_ to water through a reaction that depends on GSH as a reducing agent [24].

The impairment of GPx activity has been linked to the development of several pathological conditions, including neurodegenerative diseases like AD and Parkinson’s disease (PD), among others [125,126,127].

The selenol group within the selenocysteine residue of GPx is initially oxidized by H_2_O_2_ or other reactive oxidants, leading to the formation of a selenenic acid intermediate (GPx-SeOH). This intermediate is subsequently recycled back to its active selenol form through a two-step mechanism. First, GPx-SeOH reacts with GSH to form a selenenyl–sulfide intermediate (GPx-SeSG). Then, a second molecule of GSH reduces GPx-SeSG, regenerating the active selenol group. However, under conditions of elevated oxidative stress or limited GSH availability, GPx-SeOH can undergo further oxidation, resulting in the formation of seleninic acid (GPx-SeO_2_H), which may compromise enzyme activity [18]. Among the GPxs implicated in aging, Gpx4, also known as phospholipid hydroperoxide (PHGPx), plays a particularly prominent role. Notably, GPx4 possesses the unique ability to utilize thiol-protein as an alternative reducing agent under conditions where GSH availability is compromised [124]. GPx4 appears to exert an anti-inflammatory effect. In human dermal fibroblasts engineered to overexpress GPx4, levels of phospholipid hydroperoxides are significantly diminished. Furthermore, upon exposure to exogenous phosphatidylcholine hydroperoxides or UVA irradiation, these cells show a notable suppression in the activation of the pro-inflammatory transcription factor NF-kB and a corresponding reduction in interleukin-6 (IL-6) production [128]. GPx4 is particularly notable for its essential role in suppressing ferroptosis, which is a regulated form of cell death driven by lipid peroxidation [129,130]. Ferroptosis is a distinct type of regulated cell death characterized by a significant reduction in GPx4 activity. Unlike apoptosis or necrosis, it exhibits unique morphological and biochemical traits. This process is primarily driven by the iron-dependent accumulation of lipid peroxides [131], triggered by two key events: a reduction in GPx4 enzymatic activity, potentially accompanied by intracellular GSH depletion, and the activation of 12,15-lipoxygenase [132]. Ultimately, the non-enzymatic autoxidation of lipids acts as the terminal cytotoxic event driving ferroptosis. A growing body of evidence indicates that ferroptosis contributes to numerous pathophysiological conditions, positioning GPx4 as a promising target for future therapeutic development [133]. GPx2 is predominantly found in the gastrointestinal tract, where it functions to neutralize both inorganic and organic peroxide compounds. It is involved in regulating cellular differentiation pathways and preserving the integrity of the mucosal environment [134]. GPx3 functions as a key enzyme in the cellular antioxidant defense system [135]. Age-dependent decline in its activity has been linked to a heightened susceptibility to cardiovascular complications [136]. The presence of GPx6 has been identified in both embryonic tissues and the olfactory epithelium [18]. Its simultaneous mutation alongside GPx1, GPx2, and GPx7 leads to reduced lifespan in *Caenorhabditis elegans* [137].

Additionally, selenocysteine is essential for the enzymatic activity of TrxR, serving as a key element within its active site [138].

TrxRs are enzymes belonging to the oxidoreductase family that, together with thioredoxin (Trx), form the primary cellular system responsible for disulfide bond reduction. In mammals, this enzyme family includes three isoforms, all of which incorporate Sec within their structure, specifically in the COOH-terminal penultimate position [24].

The primary biological function of TR1 is to catalyze the reduction of Trx1 using nicotinamide adenine dinucleotide phosphate (NADPH) as an electron donor. Trx1, as a key disulfide reductase within the cell, plays a central role in various cellular processes, including protection against oxidative stress, the modulation of transcription factor activity, and the regulation of programmed cell death (apoptosis) [139,140]. Moreover, it acts as a reducing agent for various redox-dependent enzymes, such as ribonucleotide reductases, peroxiredoxins, and methionine sulfoxide reductases [141].

SELENOP contains the 21st amino acid, selenocysteine [122]. SELENOP is the only human selenoprotein known to incorporate as many as ten selenocysteine residues within its structure and is recognized as the primary carrier of selenium in mammals [142]. SELENOP is crucial for delivering selenium to the brain, where it supports the synthesis of vital selenoproteins [109], like GPx and TrxR [143]. Currently, SELENOP is regarded as a key biomarker reflecting the adequacy of selenium nutritional status [39,144]. However, SELENOP serves additional functions beyond selenium delivery [145]. In fact, it possesses a redox-active domain and demonstrates notable antioxidant activity [146,147]. SELENOP has been shown, through studies in transgenic mice, to influence brain selenium levels by participating in redox homeostasis. The genetic deletion of SELENOP or its receptor, apolipoprotein E receptor type II (ApoER2), results in a marked reduction in selenium content in the brain [148,149], which leads to progressive neurological dysfunction and neuronal degeneration [150]. Moreover, SELENOP gene expression has been shown to rise progressively with advancing age [151], suggesting a heightened requirement for selenium in aging individuals [150]. Furthermore, other selenoproteins involved in the aging process should also be mentioned.

SELENOK resides within the endoplasmic reticulum and plays a role in quality control processes by participating in the clearance of misfolded proteins through ER-associated degradation pathways. It is believed to play a role in safeguarding cells against apoptosis triggered by ER stress. Experimental evidence from murine models highlights its involvement in regulating calcium signaling within immune cells and facilitating robust immune responses [152]. SELENOR facilitates the conversion of methionine-R-sulfoxides back to methionine, contributing to cellular defense mechanisms against oxidative damage and supporting the restoration of oxidatively modified proteins [153]. Its expression levels decline as cells undergo replicative aging [154]. SELENOW has Trx-like function [155]. It modulates the maturation of osteoclasts and helps prevent bone density loss associated with osteoporosis [156]. DIO1 is involved in the regulation of thyroid hormone, and it plays a crucial role in regulating thyroid activity [157]. DIO2 is highly expressed in the brain and thyroid, where it facilitates the conversion of the pro-hormone thyroxine into the active thyroid hormone [158]. Moreover, it is essential for controlling thyroid hormone activity, and it is associated with elevated production of triiodothyronine (T3) in the thyroid gland [159]. DIO3 has been demonstrated to facilitate the conversion of thyroid hormones into inactive metabolites by removing iodine atoms from the inner ring of thyroxine and triiodothyronine [157]. It has a significant ability to reduce the impact of tissue dysfunction in human thyroid diseases [160]. SELENOF, also known as SEL 15, participates in ensuring the proper folding of glycoproteins [161]. SEL15 plays a vital role in glycoprotein folding and maintaining redox balance; cells lacking SEL15 show misfolding of lens proteins [161]. SELENOM’s precise role remains unclear, but it has been associated with the development of neurodegenerative disorders [162]. It is also linked to the regulation and support of oocyte development and maturation [163]. SELENON is situated in the ER, where it functions as a calcium-sensing protein [164]. It helps safeguard cells from oxidative stress by regulating calcium balance through redox mechanisms. Certain mutations are associated with early-onset muscle disorders [164]. SELENOS is present in the endoplasmic reticulum, where it influences and regulates the folding of proteins [165]. It also contributes to the control of lipid storage and the regulation of insulin activity [166]. SPS2 acts as a source of selenium in the synthesis of selenocysteine in mammals [167]. A lack of it worsens the progression of osteoarthritis [168]. SELENOH exhibits oxidoreductase activity, and it is involved in preventing apoptotic cell death, thereby contributing to neuronal protection from UVB-induced damage [169]. It participates in preventing cellular aging by regulating redox balance and genome stability. It also supports mitochondrial activity and the formation of new mitochondria [169]. SELENOI plays an essential role in phosphatidylethanolamine synthesis by catalyzing the conversion of CDP-ethanolamine and diacylglycerol into phosphoethanolamine [170]. In addition, it is vital for mouse embryonic development [171]. SELENOO has been demonstrated to be involved in the AMPylation of bacterial proteins [172]. It is involved in supporting mitochondrial function following selenomethionine treatment in mouse models of AD [173]. SELENOT possesses a Trx-like configuration and exhibits oxidoreductase function [174]. It also safeguards dopaminergic neurons from oxidative damage and early apoptosis [175]. SELENOV is mainly expressed in the testis, featuring a Trx-like structure, and is likely involved in redox activities [176]. It provides defense against reactive oxygen and nitrogen molecules [177].

Given their central involvement in redox signaling and protective enzymatic systems, cysteine and selenocysteine emerge as integral factors in the complex molecular network that sustains healthy aging and counteracts age-related dysfunction.

## 9. Role of Intracellular Glutathione in Cellular Aging

Glutathione is synthesized in the cytosol, but its production involves the coordinated activity of multiple organelles, including the endoplasmic reticulum, mitochondria, nuclear matrix, and peroxisomes [178,179]. Cysteine availability and the activity of the enzyme glutamate–cysteine ligase (GCL) are the primary factors regulating GSH synthesis. GCL consists of two subunits: a catalytic component (GCLC) and a regulatory modifier subunit (GCLM). This enzyme drives the first step of de novo GSH biosynthesis by combining glutamate and cysteine to form γ-glutamyl–cysteine, a bond that imparts significant structural stability to the GSH molecule. In the subsequent step, glutathione synthase (GS) catalyzes the addition of glycine to γ-glutamyl–cysteine, resulting in the formation of GSH [120,180,181,182,183,184]. Because of the stability provided by the bond between glutamate and cysteine, GSH can frequently participate in redox cycling, with its degradation tightly controlled by specific enzymes. One such enzyme, γ-glutamyl transpeptidase (GGT), breaks the γ-carboxyl linkage in glutamate, resulting in the production of the dipeptide cysteinyl–glycine (Cys-Gly) [178,179,181].

The reactive thiol group enables GSH to efficiently neutralize ROS and reactive nitrogen species (RNS) [180], thereby maintaining cellular redox homeostasis across all mammalian cells [120]. When reduced GSH is oxidized, two GSH molecules combine through a disulfide bond, resulting in the formation of oxidized glutathione (GSSG) [185]. Reduced GSH and its oxidized disulfide counterpart can be converted into each other, with GSH typically representing the most abundant form under physiological conditions [184]. GSH operates in conjunction with other redox-active molecules, such as NADPH, to regulate and preserve the redox balance within cells [186]. To be more specific, during the detoxification of peroxides, the oxidation of reduced glutathione into its disulfide form is catalyzed by the enzyme GPx, while the enzyme GR is involved in the regeneration of GSH from GSSG to maintain the intracellular redox balance [187,188].

Biochemical research on aging-related alterations in antioxidant defenses has revealed significant shifts in GSH levels within the nervous system over time [189]. These changes are frequently linked to cognitive and functional declines in the central nervous system (CNS), often due to neuronal loss, which contributes to cognitive deterioration with age. ROS, especially oxygen-derived radicals, are key contributors to both age-related cellular changes and the development of various neurodegenerative disorders. Recent investigations in rodent models have shown that aging alters GSH balance across various tissues. Specifically, they observed an age-related decline in GSH levels in all the tissues examined. This reduction was linked to decreased expression of the enzymes GS and GCL. Interestingly, this occurred without significant changes in the levels of GGT, GR, or GSSG or cysteine availability, suggesting that the drop in GSH was not due to increased oxidative stress or altered degradation [190].

The weakening of neuronal antioxidant defenses, particularly the glutathione system, contributes to heightened neuroinflammatory responses and plays a key role in the development of neurodegenerative diseases. In amyotrophic lateral sclerosis (ALS), dysfunction of the GSH-mediated antioxidant system leads to increased neuronal toxicity and accelerates disease progression. Similarly, in PD, both neurodegeneration and neuroinflammation have been closely linked to glutathione depletion [191,192,193].

These findings indicate that genetic variations in key enzymes of the glutathione antioxidant system, which regulate oxidative and nitrosative stress over the course of life, likely play a crucial role in shaping the trajectory of normal aging, especially within the central nervous system. Given that glutathione synthesis and function heavily depend on the availability of cysteine as a precursor and on selenocysteine-containing enzymes such as glutathione peroxidases, this system represents a central hub in redox homeostasis. As such, investigating the glutathione network, along with its essential components like cysteine and selenocysteine, will continue to be a priority in aging and neurobiology research for the future [182].

## 10. Therapeutic Approaches and Clinical Implications

Aging is one of the most intricate biological processes, with numerous theories proposed to explain its mechanisms. The most widely accepted is the free radical theory, which posits that the gradual accumulation of oxidative damage to cellular macromolecules by reactive oxygen species is a primary driver of the normal aging process [194].

Over the years, this theory has undergone ongoing investigation and refinement. Currently, free radicals are understood as ROS and RNS that target and damage macromolecules. These reactive intermediates arise as natural byproducts of cellular metabolism, with approximately 1–2% of the oxygen consumed during mitochondrial respiration being converted into reactive oxygen species [195].

According to the free radical theory of aging, considerable research has focused on how antioxidants may reduce cellular oxidative damage and influence lifespan and age-related changes. Dietary supplementation with resveratrol has been shown to prolong lifespan and improve physiological functions associated with aging in several model organisms [196,197]. Vitamin E (α- and γ-tocopherol), another widely recognized antioxidant, has been shown to prolong lifespan and delay age-associated gene expression changes in the brain and muscles of mice [198].

A recent investigation showed that NAC, a potent antioxidant derived from cysteine, promotes an increased lifespan and enhances resilience to environmental stress [20]. NAC is a derivative of the sulfur-containing amino acid L-cysteine, characterized by the presence of an acetyl group bonded to its nitrogen atom. It exhibits potent antioxidant properties, and it is known for its hepatoprotective effects [199]. Interestingly, NAC supplementation enhances cellular defense against oxidative stress induced by ROS by boosting intracellular glutathione concentrations [200]. Another study demonstrated that NAC enhanced both enzymatic and non-enzymatic antioxidant defenses, including superoxide dismutase (SOD), catalase, GSH, and total thiol content (T-SH), while significantly reducing levels of pro-oxidant markers such as protein carbonyls, advanced oxidation protein products (AOPP), ROS, and malondialdehyde (MDA) in aged rats. Additionally, NAC supplementation downregulated the expression of pro-inflammatory cytokines (TNF-α, IL-1β, and IL-6) and upregulated genes associated with aging (such as sirtuin-1) and neuroprotection (neuroglobin, synapsin-I, and MBP-2). These findings further support the neuroprotective role of NAC and highlight its potential therapeutic value in managing age-related neurodegenerative disorders [19].

In older adults, GSH levels tend to decline with age, contributing to increased oxidative stress and cellular dysfunction. A recent study investigated the effects of supplementation with GlyNAC, a combination of glycine and N-acetylcysteine (a source of cysteine), both of which are precursors of GSH. The results showed that this intervention restored GSH levels, reduced oxidative stress, and improved mitochondrial function [21].

Ebselen, a selenoorganic compound, was initially designed to function as a GPx mimic, utilizing GSH to neutralize ROS [201]. Ebselen’s GPx-like activity has been thoroughly investigated. Its selenium–nitrogen (Se-N) bond is easily broken by thiol-containing compounds, producing the active selenol form (EbSeH). This selenol reduces hydrogen and lipid peroxides, resulting in the formation of a selenenic acid intermediate (EbSeOH). The selenenic acid is then recycled back to the selenol state through a two-step reaction involving two GSH molecules, which first form a selenenyl–sulfide intermediate (EbSeSG) before undergoing a full reduction [202]. Given its antioxidant capabilities, ebselen is able to reduce oxidative stress and protect cells from damage induced by free radicals [25,203].

A recent study using mouse hippocampal neuronal HT22 cells and a murine model investigated the protective effects of L-cysteine against oxidative stress. The pro-oxidant compound 2,3-dimethoxy-1,4-naphthoquinone (DMNQ) was used to induce ROS-mediated apoptosis in vitro, while L-buthionine sulfoximine (BSO), an inhibitor of GSH synthesis, was administered in vivo to reduce GSH levels and induce cognitive deficits. Administration with L-cysteine was shown to mitigate ROS accumulation, restore GSH concentrations, and prevent mitochondrial damage and apoptosis in neuronal cells. In mice, oral supplementation with L-cysteine also alleviated BSO-induced memory and cognitive impairments. These findings highlight L-cysteine’s potential as a protective agent, capable of restoring redox balance and mitigating oxidative damage, even when glutathione synthesis is compromised [17]. Although L-cysteine is a less well-characterized excitotoxin, it may play a significant role in neurodegenerative diseases. Its levels rise in the brain during ischemia, and they are elevated in the plasma of patients with certain neurodegenerative disorders. Several mechanisms may underlie its neurotoxicity, particularly its interaction with glutamatergic neurotransmission. L-cysteine affects both pre- and post-synaptic components by enhancing the production of excitatory agents via cysteine dioxygenase and increasing NMDA receptor sensitivity. These effects reduce the natural inhibition of NMDA ion channels by Zn^2+^, Mg^2+^, and disulfide bonds, allowing excessive Ca^2+^ influx into neurons, which is ultimately toxic. Additionally, L-cysteine may promote extracellular glutamate accumulation, activate metabotropic glutamate receptors, and lead to further NMDA receptor activation through protein kinase C signaling. Non-NMDA receptor activity may further depolarize neurons, relieving the Mg^2+^ block. In the presence of catecholamines, L-cysteine-derived cysteinyl-catechols can inhibit mitochondrial respiration, increasing neuronal vulnerability to excitotoxic damage [204]. Therefore, excessive or unregulated supplementation with L-cysteine may carry significant neurotoxic risks, particularly in individuals predisposed to excitotoxic or neurodegenerative conditions.

Additional research suggests that S-carboxymethylcysteine, beyond its mucolytic properties, may exert neuroprotective effects by scavenging free radicals and mitigating oxidative stress and mitochondrial dysfunction. These properties support its potential role in attenuating age-related neuronal decline and slowing the progression of neurodegenerative disorders such as PD [205].

An additional aspect worth exploring is selenium supplementation, particularly in the context of gut microbiota dysregulation, one of the recognized hallmarks of aging. Selenium has been shown to influence the composition of the gut microbiota by promoting the proliferation of beneficial bacterial strains such as *Lactobacillus* and *Bifidobacterium*. These microbial shifts have been associated with reduced AD pathology, suggesting that selenium may exert indirect neuroprotective effects through the gut–brain axis. Furthermore, experimental models of AD have demonstrated that selenium supplementation can reduce amyloid-β accumulation, mitigate tau-related neurodegeneration, enhance synaptic plasticity, promote neurogenesis, and improve cognitive function [206].

The expression of selenoproteins is influenced by selenium supplementation in a hierarchical fashion. Among them, GPX4 is considered to have higher priority in expression, whereas GPX1 demonstrates greater responsiveness to changes in selenium levels, showing significant variation under both deficient and supplemented conditions [207,208]. Individuals exhibit variability in selenium metabolism, as well as in their response to supplementation, which is largely attributed to genetic polymorphisms within selenoprotein-coding genes [209,210]. In particular, functional single-nucleotide polymorphisms (SNPs) in the genes GPX1, GPX4, and SELENOP, which are responsible for encoding the respective selenoproteins, have been associated with altered blood selenium concentrations or changes in selenoprotein expression following supplementation [203,211,212]. The GPX1/rs1050450 variant involves a substitution of proline with leucine at position 197, a change that has been linked to diminished enzymatic function and increased levels of DNA damage [210,213,214,215]. The GPX1/rs1050450 polymorphism has also been linked to an elevated risk associated with various diseases, such as peripheral neuropathy [216]. The GPX4/rs713041 variant influences the interaction of proteins with the 3′-untranslated region, which is located near the selenocysteine insertion sequence that is essential for the synthesis of selenoproteins [212,217], and it has been linked to reduced activity of GPX enzymes [218]. The SELENOP/rs3877899 variant results in an alanine to threonine substitution at position 234 of the protein, while SELENOP/rs7579 is situated within the 3′-untranslated region of the SELENOP mRNA, where it involves a guanine to adenine nucleotide change [211]. Both SELENOP variants influence selenium metabolism by modifying the relative abundance of the 60 kDa isoform of SELENOP [219]. These polymorphisms have been reported to influence blood levels of selenium and SELENOP following selenium supplementation [211]. Overall, these findings suggest that these polymorphisms may play a regulatory role in modulating the expression of their respective selenoproteins. Nevertheless, it remains uncertain whether such effects are observed at selenium intake levels that are achievable through a typical diet [220].

Although most findings are derived from preclinical models, several clinical trials have explored the potential of NAC and selenium supplementation in human conditions related to aging, including neurodegenerative diseases and cardiovascular disorders. However, the evidence remains heterogeneous across different aging-related conditions, highlighting the need for more standardized and large-scale trials to confirm efficacy and safety [221,222].

Beyond its redox roles, cysteine has also been implicated in broader physiological processes. Comparative studies across species have highlighted an inverse relationship between the cysteine content in mitochondrially encoded proteins and lifespan across animal species, suggesting that higher levels of this sulfur-containing amino acid may be associated with reduced longevity. This observation provides intriguing insights into the potential role of mitochondrial protein composition in aging processes [223].

Despite the widespread use of antioxidants to counteract oxidative damage, their chronic supplementation remains a topic of debate. The recent literature has highlighted that ROS are not only damaging byproducts of metabolism but also play essential roles in physiological signaling, defense mechanisms, and cellular adaptation, as previously described in the introduction. This dual role suggests that excessive antioxidant use may disrupt essential ROS-mediated signaling pathways. While preclinical studies consistently report the positive effects of antioxidants, findings from clinical trials have been more variable and inconsistent. This discrepancy is often attributed to interindividual biological heterogeneity and differences in experimental design between animal models and human studies. Furthermore, the adverse effects of antioxidants, unlike the extensively studied role of ROS, have received relatively limited attention in research. A clearer understanding of antioxidant actions, which may range from direct ROS scavenging to the inhibition of ROS-producing enzymes, is essential. These observations underscore the importance of a balanced and individualized approach to redox modulation, avoiding indiscriminate antioxidant use and considering the context-specific roles of ROS in health and disease [224]. The following table (Table 1) summarizes the therapeutic approaches and clinical trials described in this review.

An important aspect that needs careful evaluation is the potential long-term toxicity and bioavailability of supplemented selenocysteine and cysteine. A recent review analyzing the chronic intake of various selenium compounds, including selenocysteine and selenomethionine, has indicated that adverse health effects may arise at doses of approximately 2–3 µg of selenium per kilogram of body weight per day. This level represents the upper safety threshold for long-term exposure, beyond which there is evidence of increased risk for mortality, type 2 diabetes, and toxic effects on the hepatic, cardiovascular, and reproductive systems [225]. Interestingly, organic selenium compounds typically show greater absorption and lower toxicity compared to their inorganic counterparts, highlighting the importance of selecting the appropriate chemical form for supplementation [226]. Similarly, a study investigating prolonged dietary cysteine enrichment in aged rodents found that although elevated intake led to increased concentrations of free cysteine and glutathione in hepatic and intestinal tissues, it did not yield measurable benefits in terms of systemic inflammation, survival rates, or the preservation of lean body mass. Moreover, only cysteine-supplemented animals maintained their food intake over time, whereas control animals showed a decline, suggesting that the physiological advantages of high-dose cysteine supplementation in otherwise healthy aging organisms may be limited [227]. Taken together, these observations highlight the importance of evaluating the dose, chemical form, duration of use, and tissue-specific absorption when considering supplementations based on cysteine or selenocysteine.

The meta-analysis titled “Meta-Analysis of randomized clinical trials evaluating effectiveness of a multivitamin supplementation against oxidative stress in healthy subject” systematically reviewed randomized clinical trials, assessing whether multivitamin supplementation effectively reduces oxidative stress biomarkers in healthy individuals [228]. While multivitamins and antioxidant supplements are often promoted for their potential to mitigate oxidative damage associated with aging and chronic diseases, this meta-analysis revealed mixed and inconclusive results. Although some trials showed modest improvements in specific oxidative stress markers, the overall pooled effect was not consistently significant across studies. This suggests that the antioxidant effects of multivitamin supplementation might be limited or variable in healthy populations. Importantly, the meta-analytic approach strengthens these conclusions by integrating data from multiple trials, thus providing a more comprehensive and statistically robust evaluation than individual studies alone. However, the study also highlights significant heterogeneity among trials in terms of supplement composition, dosage, and duration, as well as participant characteristics, which complicate the conclusions. These findings underscore that antioxidant supplementation can vary significantly depending on individual factors and context. It is crucial to consider that excessive or imbalanced antioxidant intake might disrupt endogenous redox homeostasis, possibly leading to unexpected consequences. Therefore, this meta-analysis indicates that while multivitamin supplementation may exert some antioxidant effects, the evidence does not support universal efficacy or safety, particularly in healthy subjects. These results highlight the importance of continued research into how antioxidant supplementation can be customized to individual needs, with a focus on safe dosing and sustained health benefits over time.

## 11. Discussion

The intricate interplay among oxidative stress, redox signaling, and the aging process underscores the central role of redox-active amino acids, such as cysteine and selenocysteine, in maintaining cellular homeostasis [18,26]. As shown throughout this review, both amino acids play essential functions in redox regulation, antioxidant defense, and mitochondrial integrity processes that are deeply implicated in age-associated physiological decline and the pathogenesis of chronic diseases.

Cysteine plays a pivotal role as a precursor of GSH, which is the most abundant intracellular antioxidant [229]. Its availability directly influences GSH synthesis, and consequently, the cell’s capacity to neutralize ROS. Notably, age-related reductions in GSH levels, particularly in the CNS, have been associated with impaired neuronal function and increased susceptibility to neurodegeneration [182]. Experimental studies have demonstrated that supplementation with cysteine precursors, such as NAC, or GlyNAC (a combination of glycine and NAC), can restore intracellular GSH pools, mitigate oxidative stress, and improve mitochondrial function, thereby potentially counteracting several hallmarks of aging [20,21,199,200].

Selenocysteine, on the other hand, is uniquely incorporated into selenoproteins via a highly specialized translational machinery. As the active site residue of enzymes such as GPx and TrxR, selenocysteine enhances catalytic efficiency in detoxifying peroxides and maintaining redox balance [230]. The functional integrity of these selenoenzymes is critical for the neutralization of ROS and the preservation of redox signaling, especially under conditions of elevated oxidative stress observed in aging tissues [231].

Experimental and clinical studies support the potential therapeutic relevance of modulating cysteine and selenocysteine levels in the aging process. While NAC and selenium supplementation have shown beneficial effects in preclinical models of neurodegeneration, cardiovascular dysfunction, and metabolic disorders, clinical trials have yielded more heterogeneous results. Such variability is likely influenced by differences in dosing, population characteristics, and trial design. Furthermore, emerging evidence suggests that excessive antioxidant supplementation may interfere with essential ROS-dependent signaling pathways, raising concerns about the long-term use of such supplements in humans.

Importantly, the biological effects of ROS are dose-dependent; while high concentrations are damaging, physiological levels are crucial for signaling and adaptation [44]. This dual nature of ROS emphasizes the need for a blurred approach to redox modulation, one that considers the balance between damage prevention and the preservation of essential redox-mediated processes.

Recent advances in computational biology offer promising tools to address these complexities. Machine learning models capable of predicting redox imbalances based on multi-omics aging profiles could provide a personalized framework for optimizing cysteine and selenocysteine supplementation strategies. By integrating genetic, metabolic, and environmental data, such models may help identify individuals most likely to benefit from targeted redox interventions while minimizing the risks associated with indiscriminate antioxidant use. This precision approach could enhance the therapeutic potential of redox-active amino acids in aging and age-related disease contexts.

## 12. Conclusions

Aging is a multifactorial biological process characterized by the progressive loss of physiological integrity, in which oxidative stress and redox imbalance play central roles [232]. In this context, the thiol-containing amino acids cysteine and selenocysteine emerge as critical components of the antioxidant defense system, contributing to cellular protection through distinct yet complementary mechanisms [233].

Cysteine supports redox homeostasis primarily as a precursor for GSH, the most abundant intracellular antioxidant [229]. The availability of cysteine directly influences GSH synthesis and, consequently, the ability of cells to counteract oxidative insults. Its role becomes particularly relevant during aging, when GSH levels decline, compromising detoxification capacity and mitochondrial function [234].

Selenocysteine, which is uniquely incorporated into a subset of selenoproteins through a specialized translational mechanism, provides catalytic efficiency in redox-active enzymes such as GPx and TrxR. These enzymes are essential for detoxifying peroxides and maintaining redox signaling, especially in tissues susceptible to oxidative damage during aging [124]. Evidence from both preclinical models and clinical observations supports the potential of cysteine and selenium supplementation in mitigating oxidative stress, improving mitochondrial function, and possibly slowing age-associated functional decline [235]. However, the clinical translation of these findings remains complex, and further research is needed to define optimal dosages, treatment windows, and long-term effects.

In summary, cysteine and selenocysteine represent promising targets in the modulation of redox homeostasis during aging, offering avenues for preventive and therapeutic strategies against age-related diseases. Their integration into personalized interventions aimed at preserving redox balance may contribute to healthier aging and extended health span.

## 13. Future Perspectives

Future directions in aging research underscore that the role of thiol-containing amino acids extends beyond simple antioxidant activity. Cysteine and selenocysteine play integral roles in regulating redox homeostasis, cellular resilience, and stress responses [18]. Future research should focus on refining antioxidant strategies, elucidating the dose- and context-specific roles of these amino acids, and developing targeted interventions that enhance redox capacity without impairing physiological redox signaling. A deeper understanding of these mechanisms holds promise for the development of more effective interventions aimed at mitigating the functional decline associated with the aging process. Harnessing the full potential of cysteine and selenocysteine in redox biology may pave the way for innovative, personalized strategies for promoting resilience against age-related decline and extending the healthy human lifespan.

## 14. Patents

Not applicable.

## Figures and Tables

**Figure 1 biomolecules-15-01115-f001:**
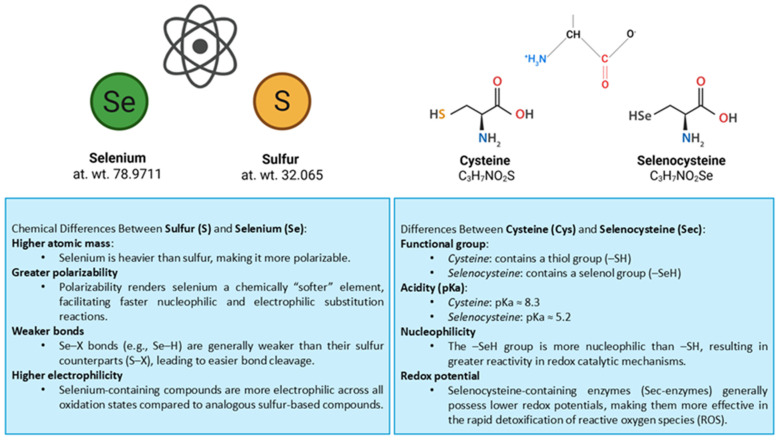
Main differences between sulfur and selenium atoms at the chemical level and the molecular differences between cysteine and selenocysteine amino acids.

**Table 1 biomolecules-15-01115-t001:** Main therapeutic approaches and clinical trials related to oxidative stress and age-related dysfunctions. GPx: glutathione peroxidases; GSH: glutathione; NAC: N-acetyl-L-cysteine; PD: Parkinson’s disease; ROS: reactive oxygen species; SIRT1: sirtuin-1.

Molecule	Structure	Mechanism of Action	Observed Effects	Models
Resveratrol	Natural Polyphenol	Activates sirtuins and antioxidants	Extends lifespan; improves physiological functions.	Preclinical model: Saccharomyces Cerevisiae [197], Caenorhabditis elegans, Drosophila melanogaster [196]
Vitamin E (α- and γ-Tocopherol)	Lipid-soluble vitamin	Lipid-targeted antioxidant	Extends lifespan, delays age-associated gene expression in the brain and muscles	Preclinical model mice B6C3F [198]
N-Acetyl-L-Cysteine (NAC)	Acetylated derivative molecule of cysteine	Enhances GSH levels and ROS scavenger and modulates inflammatory cytokines	Strong antioxidant; neuroprotective; reduces pro-oxidants; upregulates SIRT1 and neuroprotective genes	Preclinical trials: C. elegans [20], HepG2 cells [199], Human dermal fibroblast primary cells [200], Wister rats [19]. Limited clinical trials [221,222]
Glycine and N-Acetylcysteine (GlyNAC)	Combined molecules of glycine and N-acetylcysteine	Precursors of GSH; restores glutathione synthesis	Restores GSH levels; improves mitochondrial function; reduces oxidative stress	Preliminary human studies [21]
Ebselen	Organoselenic compound	Mimics GPx; neutralizes peroxides via active selenol form (EbSeH)	Reduces oxidative damage; mimics GPx function; regenerates via GSH-dependent redox cycle	Preclinical in vitro studies [201]. Preclinical models: rodents and dogs [202], rodents [203]
L-Cysteine	Sulfur-containing amino acid	GSH precursor; direct antioxidant; mitochondrial protective effects	Prevents ROS-induced apoptosis; restores GSH; improves cognitive deficits in oxidative stress models	Preclinical in vitro HT22 cells and mouse studies [17]
Selenium	Compounds containing selenium	Increases expression or function of selenoproteins; It modulates gut microbiota biodiversity	Improved microbiota composition	Animal models of AD [206]
S-Carboxymethylcysteine	Derivative molecule of cysteine	Free radical scavenger; protects against mitochondrial dysfunction	Neuroprotective; delays age-related neuronal degeneration, including in PD models	Preclinical in vitro SH-SY5Y cell line [205]

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
