# Peer review of "Exploring the Antioxidant Roles of Cysteine and Selenocysteine in Cellular Aging and Redox Regulation"

_biomolecules, 2025, doi:10.3390/biom15081115_

Round 1

Reviewer 1 Report

Comments and Suggestions for Authors

Article of great interest and relevance. It is necessary to review the article regarding the use of abbreviations and to abbreviate them in the text only the first time they appear. There are other minor writing errors, as evidenced in the attached PDF. Table 1 should include a legend with the description of all abbreviations used in the text. The bibliography should be carefully reviewed and updated with more recent studies where appropriate. A simple assessment of the potential long-term toxicity and bioavailability of the proposed substances as sources of cysteine or selenocysteine is also required.

Author Response

Reviewer 1

Article of great interest and relevance. It is necessary to review the article regarding the use of abbreviations and to abbreviate them in the text only the first time they appear. There are other minor writing errors, as evidenced in the attached PDF. Table 1 should include a legend with the description of all abbreviations used in the text. The bibliography should be carefully reviewed and updated with more recent studies where appropriate. A simple assessment of the potential long-term toxicity and bioavailability of the proposed substances as sources of cysteine or selenocysteine is also required.

Response: We appreciate the Reviewer’s comments. Thank you very much for your constructive feedback and for the careful reading of my manuscript. In response to your observations, we have undertaken a thorough revision of the text. Specifically, we have standardized the use of abbreviations, ensuring that each is clearly defined upon first mention and used consistently throughout the manuscript. A comprehensive legend has been added to Table 1, detailing all abbreviations used, as recommended. The reference list has been carefully reviewed and updated to include several recent and relevant studies, to strengthen the scientific foundation of the review. Finally, a dedicated section has been added to address the potential long-term toxicity and bioavailability of cysteine and selenocysteine supplementation, providing a more complete and balanced discussion of their therapeutic implications. We sincerely appreciate your insightful suggestions, which have contributed to improving the clarity and scientific rigor of the manuscript.

Reviewer 2 Report

Comments and Suggestions for Authors

The manuscript offers a broad and comprehensive overview of oxidative stress, redox biology, aging mechanisms, and the distinct roles of cysteine and selenocysteine. While informative, the review would benefit from refinement and deeper integration of emerging research. Below are specific comments:

1. Several sections repeat similar content (e.g., redox buffering via GSH and NAC’s role), which reduces conciseness and weakens the manuscript’s novelty.

2. While the biochemical background is strong, mechanistic insights into how redox signaling influences gene regulation and epigenetic modifications during aging are lacking.

3. The influence of SNPs in selenoprotein genes (e.g., SELENOP, GPX1) on aging or supplementation response is not addressed. Additionally, although epigenetic drift is listed as a hallmark of aging, the paper does not explore how epigenetic regulation (e.g., DNA methylation of GSH-related genes) modulates cysteine or selenocysteine metabolism.

4. The role of gut microbiota is briefly mentioned but not linked to thiol or selenium metabolism. Recent findings show microbial regulation of selenium bioavailability and GSH synthesis.

5. The focus is limited to GPx and TrxR, overlooking other selenoproteins (e.g., selenoproteins H, W, M) with emerging roles in neuroprotection and aging. Importantly, GPX4 and other selenoproteins are central to ferroptosis, a redox-regulated cell death pathway associated with aging.

6. There’s little to no mention of high-throughput or multi-omics approaches to studying redox biology in aging (e.g., transcriptomic or proteomic studies on redox imbalance in aging, redox proteomics to monitor cysteine oxidation states, metabolomic profiling of thiol and selenium metabolism)

7. The manuscript supports antioxidant supplementation without critically addressing the mixed or negative results from large-scale clinical trials.

8. Several claims promote cysteine supplementation but fail to mention the known risks of free L-cysteine, such as auto-oxidation, neurotoxicity, and H2S-mediated cytotoxicity.

9. Emerging CRISPR-based strategies to enhance selenoprotein expression are relevant but unmentioned.

10. The paper omits discussion of Nrf2-KEAP1 modulation, a key redox regulatory pathway tied to GSH synthesis, stress resilience, and healthy aging.

11. Machine learning models predicting redox imbalances in aging could provide insights into optimal cysteine/selenocysteine supplementation strategies, an area the paper does not address.

Author Response

Reviewer 2

The manuscript offers a broad and comprehensive overview of oxidative stress, redox biology, aging mechanisms, and the distinct roles of cysteine and selenocysteine. While informative, the review would benefit from refinement and deeper integration of emerging research. Below are specific comments:

Response: We would like to express our sincere gratitude for your careful and thoughtful evaluation of our review. Your in-depth analysis and constructive observations have been invaluable in guiding the revision process. We have taken each of your suggestions into close consideration and implemented substantial improvements throughout the manuscript. Please find below a detailed point-by-point response to your comments

  1. Several sections repeat similar content (e.g., redox buffering via GSH and NAC’s role), which reduces conciseness and weakens the manuscript’s novelty.

Response: We apologize for the oversights. We have thoroughly revised the manuscript to eliminate repetitive descriptions, to improve clarity and enhance the overall conciseness and originality of the review.

  1. While the biochemical background is strong, mechanistic insights into how redox signaling influences gene regulation and epigenetic modifications during aging are lacking.

Response: We totally agree with the reviewer and have expanded the section discussing how redox signaling impacts gene expression and epigenetic regulation during aging. This includes mechanisms by which ROS influence transcription factor activity, chromatin remodeling, and histone/DNA modifications, with relevant citations added.

  1. The influence of SNPs in selenoprotein genes (e.g., SELENOP, GPX1) on aging or supplementation response is not addressed. Additionally, although epigenetic drift is listed as a hallmark of aging, the paper does not explore how epigenetic regulation (e.g., DNA methylation of GSH-related genes) modulates cysteine or selenocysteine metabolism.

Response: We appreciate the reviewer’s comment. The revised manuscript now includes a discussion on how functional SNPs in selenoprotein genes such as SELENOP, GPX1 and GPX4 may influence aging trajectories and the response to selenium supplementation. Additionally, we integrated recent findings on how epigenetic mechanisms, including DNA methylation of redox-related genes, modulate cysteine and selenocysteine metabolism.

  1. The role of gut microbiota is briefly mentioned but not linked to thiol or selenium metabolism. Recent findings show microbial regulation of selenium bioavailability and GSH synthesis.

Response: We agree with the Reviewer, and we expanded the discussion on gut microbiota to include its role in selenium absorption as suggested. Emerging studies demonstrating microbial regulation of selenium bioavailability and thiol homeostasis have been incorporated to reflect the latest developments in this field.

  1. The focus is limited to GPx and TrxR, overlooking other selenoproteins (e.g., selenoproteins H, W, M) with emerging roles in neuroprotection and aging. Importantly,

GPX4 and other selenoproteins are central to ferroptosis, a redox-regulated cell death pathway associated with aging.

Response: we appreciate the Reviewer’s comment. The manuscript now covers additional selenoproteins implicated in aging and neuroprotection. I also emphasized the central role of GPX4 and its involvement in ferroptosis, a redox-sensitive form of cell death relevant to aging.

  1. There’s little to no mention of high-throughput or multi-omics approaches to studying redox biology in aging (e.g., transcriptomic or proteomic studies on redox imbalance in aging, redox proteomics to monitor cysteine oxidation states, metabolomic profiling of thiol and selenium metabolism)

Response: We totally agree with the reviewer, and we added a section which highlights the value of high-throughput approaches in aging research. These tools are presented as promising strategies to map redox imbalance and explore metabolic shifts in cysteine/selenium biology.

  1. The manuscript supports antioxidant supplementation without critically addressing the mixed or negative results from large-scale clinical trials.

Response: We appreciate the Reviewer comment and we now discussed the mixed results from large-scale antioxidant supplementation trials, including a more nuanced view of their efficacy and limitations in age-related disease prevention as suggested.

  1. Several claims promote cysteine supplementation but fail to mention the known risks of free L-cysteine, such as auto-oxidation, neurotoxicity, and H2S-mediated cytotoxicity.

Response: The revised manuscript now acknowledges and discusses the potential risks associated with free L-cysteine, such as neurotoxicity, to improve a balanced perspective on supplementation

  1. Emerging CRISPR-based strategies to enhance selenoprotein expression are relevant but unmentioned.

Response: We thank the reviewer for the suggestion, we now included recent studies exploring CRISPR/Cas9-based strategies to manipulate selenoprotein expression, including targeted disruption or enhancement of key biosynthetic genes, and discussed their relevance to aging research.

  1. The paper omits discussion of Nrf2-KEAP1 modulation, a key redox regulatory pathway tied to GSH synthesis, stress resilience, and healthy aging.

Response: A comprehensive description of the Nrf2-KEAP1 antioxidant regulatory pathway has been added as suggested.

  1. Machine learning models predicting redox imbalances in aging could provide insights into optimal cysteine/selenocysteine supplementation strategies, an area the paper does not address.

Response: Finally, we included a section on the use of machine learning models to predict redox imbalances during aging as suggested.

Round 2

Reviewer 2 Report

Comments and Suggestions for Authors

All comments have been successfully addressed by the authors. In my view, this version of the manuscript represents a significant improvement and will undoubtedly be of interest to the scientific audience.